# Anomalous Thermal Characteristics of Poly(ionic liquids) Derived from 1-Butyl-2,3-dimethyl-4-vinylimidazolium Salts

**DOI:** 10.3390/polym14020254

**Published:** 2022-01-08

**Authors:** Fan Yang, Meng Zhao, Darren Smith, Peggy Cebe, Sam Lucisano, Thomas Allston, Thomas W. Smith

**Affiliations:** 1Versick Analytics, Jersey City, NJ 07310, USA; yangf0530@gmail.com; 2BOE Technology Group, Beijing 100176, China; 15101035632@139.com; 3Walter Reed Army Institute of Research, Silver Spring, MD 20910, USA; darren.m.smith37.mil@mail.mil; 4Physics and Astronomy Department, Tufts University, Medford, MA 02155, USA; Peggy.Cebe@tufts.edu; 5Xerox Corporation, Webster, NY 14580, USA; slucisano2387@gmail.com; 6School of Chemistry & Materials Science, Rochester Institute of Technology, Rochester, NY 14623, USA; tdasch@rit.edu

**Keywords:** poly(ionic liquids), ionic liquid polymer, thermal analysis, ion-conductive polymer, vinylimidazolium monomers, ionic liquids

## Abstract

The synthesis of 1-butyl-2,3-dimethyl-4-vinylimidazolium triflate, its polymerization, and ion exchange to yield a trio of 1-butyl-2,3-dimethyl-4-vinylimidazolium polymers is described. Irrespective of the nature of the anion, substitution at the 2-position of the imidazolium moiety substantially increases the distance between the anion and cation. The methyl substituent at the 2-position also served to expose the importance of H-bonding for the attractive potential between imidazolium moiety and anions in polymers without a methyl group at the 2-position. The thermal characteristics of poly(1-butyl-2,3-dimethyl-4-vinylimidazolium) salts and corresponding poly(1-ethyl-3-methyl-4-vinylimidazolium) salts were evaluated. While the mid-point glass transition temperatures, T_g_-mid, for 1-ethyl-3-methyl-4-vinylimidazolium polymers with CF_3_SO_3_^−^, (CF_3_SO_2_)_2_N^−^ and PF_6_^−^ counterions, were 153 °C, 88 °C and 200 °C, respectively, the T_g_-mid values for 1-butyl-2,3-dimethyl-4vinylimidazolium polymers with corresponding counter-ions were tightly clustered at 98 °C, 99 °C and 84 °C, respectively. This dramatically reduced influence of the anion type on the glass transition temperature was attributed to the increased distance between the center of the anions and cations in the 1-butyl-2,3-dimethyl-4-vinylimidazolium polymer set, and minimal H-bonding interactions between the respective anions and the 1-butyl-2,3-dimethyl-4-vinylimidazolium moiety. It is believed that this is the first observation of substantial independence of the glass transition of an ionic polymer on the nature of its counterion.

## 1. Introduction

Ionic liquids are salts with low melting points (often below room temperature) and are typically composed of sulfonium, phosphonium, or ammonium (imidazolium, pyridinium, pyrrolidinium) cations paired with anions of low Lewis basicity (tetrafluoroborate (BF_4_^−^), hexafluorophosphate (PF_6_^−^), triflate (Tf^−^), trifluoromethylsulfonylimide (TFSI^−^), etc.). Today, the utility of ionic liquids in electrochemical devices [1,2] ranging from lithium-ion batteries [3,4] to fuel cells [5,6], capacitors [7,8], solar cells [9,10] and actuators [11], is being actively explored. Because of the mobility of both the anionic and cationic components of ionic liquids, the function of some devices might be improved if conventional ionic liquids are replaced by film-forming ionic liquid/polymer gel electrolytes or ionic liquid polymers in which the mobility of one or both ions is constrained.

Among poly(ionic liquids), 1-vinylimidazolium polymers were studied most extensively. A 2013 paper by Long et al. [12] that reports on the synthesis, ionic conductivity and thermal properties (glass transition temperature, T_g_, and thermal stability) of a series of 3-alkyl-1-vinylimidazolium polymer salts (Br^−^, BF_4_^−^, Tf^−^, and TFSI^−^), also provides a concise reference trail that is inclusive of much of the published work on vinylimidazolium polymers [13,14,15,16,17,18,19,20,21,22,23,24,25,26]. In the paper by Long et al. [12], the glass transition temperature of poly(3-alkyl-1-vinylimidazolium) salts, regardless of the alkyl substituent length, was reported to decrease as the anion size increased. Specifically, it was reported that the glass transition temperatures of Br^−^, BF_4_^−^, Tf^−^, and TFSI^−^ salts of the 3-ethyl-1-vinylimidazolium polymer (PEVIm^+^) were 217, 181, 145 and 56 °C, respectively. Similarly, the reported glass transition temperatures of Br^−^, BF_4_^−^, Tf^−^, and TFSI^−^ salts of the 3-butyl-1-vinylimidazolium polymer (PBVIm^+^) were 192, 148, 134, and 48 °C, respectively. Vygodskii et al. [27] found that the glass transition temperature of PEVIm^+^ salts was 19 °C when the counterion was (CN)_2_N^−^, 60 °C for (CF_3_SO_2_)_2_N^−^, 173 °C for CF_3_SO_3_^−^ and 235 °C for Br^−^. Elabd and co-workers [15] investigated the glass transition temperatures and conductivities of poly(1-(2-methacryloyloxy)ethyl-3-butylimidazolium) salts and reported that the glass transition temperatures of Br^−^, BF_4_^−^, PF_6_^−^, Tf^−^, and TFSI^−^ salts were 102, 85, 94, 64 and 7 °C, respectively. In all of these reports, the glass transition temperature was found to exhibit a significant dependence on the nature of the anion. Colby et al. [28] plotted the glass transition temperature and repeat unit molecular volume of imidazolium-based polyionic liquids, PILs, with a variety of different counterions and observed an ostensibly exponential decrease in T_g_ with increasing molecular volume. More recently, Bocharova and Sokolov, et al. reported that the glass transition in poly(ionic liquids) does not scale simply with the volume of structural units (monomer and counterion) and proposed an empirical model that includes electrostatic interactions and chain flexibility to describe T_g_ in ionic liquid polymers [29]. Decoupling of ion transport from segmental mobility was also proposed as a mechanism for the realization of enhanced ion conductivity [30,31,32].

In our research, we focused on 4-vinylimidazolium polymers [33]. The molecular structure of a pentad segment of a 4-vinylimidazolium polymer is shown in Figure 1.

As compared to the more widely studied 1-vinylimidazolium polymers [12,13,14,15,16,17,18,19,20,21,22,23,24,25,26,34,35,36,37], the pendant imidazolium group in 4-vinylimidazolium polymers can exhibit additional degrees of freedom, increased free volume, and enhanced lateral overlap between proximate imidazolium residues that may be situated 1,3 or 1,5 with respect to each other on the carbon chain. Work from our laboratory on the synthesis, thermal characteristics, and dielectric properties of poly(ionic liquids) derived from 1,3-dialkyl-4-vinylimidazolium salts presented data on the comparative thermal and dielectric characteristics of the 4-vinyl and 1-vinylimidazolium salts [33]. In that work, it was shown that the glass transition characteristics of 4-vinyl- and 1-vinylimidazolium salts are similar, with the glass transition temperatures of P4VIm^+^ [poly(1-ethyl-3-methyl-4-vinylimidazolium)], BF_4_^−^, PF_6_^−^, AsF_6_^−^, and Tf^−^ salts being higher than those of the corresponding poly(1-vinylimidazolium) salts. This difference, and the increase in T_g_ of P4VIm^+^ salts of complex fluoride ions of increasing size (BF_4_^−^, PF_6_^−^, and AsF_6_^−^), was attributed to enhanced intramolecular bridging between imidazolium moieties positioned 1,3 or 1,5 along the 4-vinylimidazolium polymer chain. The glass transitions of the 1-vinyl- and 4-vinylimidazolium polymers with TFSI^−^ and C_2_N_3_^−^ anions exhibited the lowest glass transition temperatures. The lower glass transition temperatures of the TFSI^−^ and C_2_N_3_^−^ salts were attributed to plasticization by large solvating anions, and the soft nucleophilic character of these anions allows for the association between the anion and cation over a larger distance.

In 1-vinyl- and 4-vinyl-imidazolium polymers, the hydrogen atom at the 2-position of the imidazole ring is somewhat acidic (p*K_a_* = 21–23) [38,39]. This acidic character can be the source of chemical and electrochemical instability [40]. In 2-methylimidazolium moieties, the problem is mitigated [41]. This paper describes the synthesis and polymerization of 1-butyl-2,3-dimethyl-4-vinylimidazolium triflate (2,3DM4VIm^+^ Tf^−^) (see Figure 2) and the thermal properties of polymers derived therefrom.

## 2. Materials and Methods

### 2.1. NMR

^1^H-NMR and ^19^F-NMR spectra were obtained using a Bruker DRX-300 spectrometer and a Brucker Advance III 500 spectrometer, respectively. Monomeric samples were dissolved in chloroform-d (Aldrich, St. Louis, MO, USA 99.8 atom % D, 0.05% *v*/*v* TMS) or methanol-d4 (Aldrich, St. Louis, MO, USA, 99.96 atom % D). Polymers were dissolved in dimethylsulfoxide-d6 (ACROS Organics, Geel, Belgium, 99.9 atom % D). The ^19^F NMR spectrum of poly(1-butyl-2,3-dimethyl-4-vinylimidazolium triflate) shows a single sharp peak at ~78 ppm. When fully ion-exchanged to poly(1-butyl-2,3-dimethyl-4-vinylimidazolium hexafluorophosphate), no trace of a peak at ~78 ppm is observed, as it is replaced by a doublet centered at ~75 ppm with a coupling constant of about 711 Hz. This splitting arises by virtue of coupling between ^19^F and ^31^P. These spectra are included in the Appendix A.

### 2.2. Thermal Analysis

Thermal gravimetry (TG) was carried out under a nitrogen atmosphere with a TA Instruments TGA 2050. The temperature was increased from 25–600 °C at 20 °C/min, and then held at 600 °C for 10 min in air. The thermal stabilities of the triflate and TFSI^−^ salts are similar. Significant mass loss in the TFSI^−^ salt occurs between 430 and 505 °C; that in the triflate salt occurs between 440 and 505 °C. TGA mass loss profiles for triflate and TFSI^−^ salts of 1-butyl-2,3-dimethyl-4-vinylimidazolium polymers are provided in the Appendix A.

Glass transition (T_g_) data were obtained under a nitrogen atmosphere by differential scanning calorimetry (DSC) using a TA Instruments DSC 2010 with refrigerated cooling system. All samples were prepared in an Ar-filled glove box. Polymer samples were placed in an open, hermetically sealable aluminum pan and heated at 100 °C for 15 min on the surface of a digital hot plate in the glove box. The aluminum pan was then capped and sealed. In the DSC, samples were ramped to 200 °C and then cooled to −50 °C at a rate of 20 °C/min. Between each heating and cooling cycle, samples were held for 1 min at −50 °C and 200 °C, respectively. T_g_ values are reported as mid-point glass transition temperatures, T_g_-mid. The analysis was a seven-step process: (1) heating from 22 °C to 200 °C at 20 °C/min; (2) holding for 1 min at 200 °C; (3) cooling from 200 to −50 °C at 20 °C/min; (4) holding for 1 min at −50 °C; (5) heating from −50 °C to 200 °C at 20 °C/min; (6) holding for 1 min at 200 °C; and (7) cooling from 200 °C to −50 °C at 20 °C/min. The respective polymers were subjected to at least four heating and cooling cycles and results showed that the second and third heating cycles were essentially identical to each other. Second-cycle DSC thermograms for the triflate polymer are shown in Figure 3, where plots of change in heat flow rate in 20 °C/min heating and cooling cycles are also shown. An auxiliary 40 °C/min rapid heating cycle, which served to validate the change in heat capacity at the glass transition [42], is included in Figure 3.

### 2.3. Size Exclusion Chromatography

Molecular weight and polydispersity were determined using an Agilent (Santa Clara, CA, USA) 1100 series gel permeation chromatograph with two Agilent Zorbax PSM 60-S columns (in series) (Santa Clara, CA, USA). The samples were eluted at 35 °C using N,N-dimethylformamide as the solvent. Molecular weight values reported are styrene-equivalent molecular weights based on hydrodynamic radii.

### 2.4. Reagents

Unless noted otherwise, all intermediates and reagents purchased from chemical supply houses were used as received. Urocanic acid (4-imidazole acrylic acid, 99%), butyl lithium (1.6M solution in hexane), *sec*-butyl lithium (1.4 M solution in cyclohexane), 1,1,1,3,3,3-hexamethyldisilazane (99.9%), *N,N*-Dimethylformamide (anhydrous, 99.8%), benzene (anhydrous, 99.8%), 1-bromobutane (99%), methyl trifluoromethanesulfonate (≥98%), *bis*(trifluoromethane) sulfonimide lithium salt, dimethylsulfoxide (99.6+%), ammonium hexafluorophosphate (99.99%), and 2,2’-azobisisobutyronitrile (AIBN, 98%) were obtained from Sigma-Aldrich (St. Louis, MO, USA). The latter was recrystallized from methanol prior to use. The chemicals 4-*tert*-Butylcatechol (99%), potassium *tert*-butoxide (1 M solution in THF, AcroSeal), triphenylmethyl chloride (98%), iodomethane (stabilized, 99%), methyl alcohol (reagent ACS, 99.8%), tetrahydrofuran (stabilized, 99+ %), ethanol (95%, denatured with 5% wood spirit), potassium carbonate (anhydrous, ACS reagent grade), ammonium sulfate, and sodium bicarbonate (ACS reagent grade) were obtained from ACROS Organics (Waltham, MA, USA). Ethyl acetate (AR ACS, 99.5%), hydrochloric acid (AR ACS), chloroform (AR ACS) and dichloromethane (AR ACS) were obtained from Mallinckrodt Chemicals (Dublin, Ireland). Diethyl ether (anhydrous) and acetone (certified ACS) were obtained from Fisher Scientific (Waltham, MA, USA). Magnesium sulfate (anhydrous powder) and acetic acid (glacial) were obtained from J.T. Baker. Triethylamine (99%) was obtained from Lancaster Synthesis (Windham, NH, USA). Acetonitrile (GR ACS) was obtained from EMD Chemicals.

#### 2.4.1. 4(5)-Vinylimidazole **(1)**

The chemical 4(5)-Vinylimidazole was synthesized by decarboxylation of urocanic acid. The procedure employed was analogous to that of Overberger, et al. [43]. Thus, urocanic acid (3.70 g, 26.8 mmol) was decarboxylated in vacuo (10 µm Hg) at 230 °C to yield 1.46 g (58%) of crude 4(5)-vinylimidazole.

#### 2.4.2. 1-Trityl-4-vinylimidazole **(2)**

The chemical 1-Trityl-4-vinylimidazole was synthesized by a procedure analogous to that of Schiavone, et al [44]. Thus, (10 g, 0.11mol) of crude 4(5)-vinylimidazole was reacted with triphenylmethyl chloride (32.6 g, 117 mmol) to provide a quantitative yield (36 g) of 1-trityl-4-vinylimidazole.

#### 2.4.3. 1-Trityl-2-methyl-4-vinylimidazole **(3)**

The chemical 1-Trityl-2-methyl-4-vinylimidazole was synthesized by a procedure analogous to that alluded to by Schiavone et al. [44], in which 1-trityl-4-vinylimidazole (12 g, 36 mmol) in dry tetrahydrofuran (600 mL, distilled from sodium benzophenone ketyl) under argon was lithiated (25.5 mL, 36 mmol, 1.4 M *sec*-butyl lithium in cyclohexane). The reaction mixture, which turned deep red upon addition of *sec*-butyl lithium, was stirred in an ice water bath for 15 min and held for 2 h at room temperature prior to alkylation with iodomethane (3 mL, 0.05 mol). The resultant yellow solution was stirred at room temperature for 30 min and quenched with distilled water (100 mL). THF was removed by rotary evaporation and the aqueous residue was exhaustively extracted with chloroform. The chloroform extract was dried over magnesium sulfate and filtered, prior to removal of the solvent in vacuo, leaving a yellow solid, 1-trityl-2-methyl-4-vinylimidazole. Crude yield = 9.3 g, 74%.

#### 2.4.4. 2-Methyl-4-vinylimidazole **(4)**

Crude 1-trityl-2-methyl-4-vinylimidazole (5.7 g, 16.4 mmol) was charged to a 500 mL single-neck, round-bottom flask equipped with a reflux condenser. An amount of 5% acetic acid–methanol (200 mL) was added to the flask and the reaction mixture was refluxed for 40 min, heating in a 75 °C oil bath. The solvent was removed by rotary evaporation and 50 mL of distilled water was added. A white precipitate formed immediately. The precipitate was centrifuged down and the water layer was decanted to another 200 mL single-neck, round-bottom flask. Residual water was removed in vacuo to yield 2-methyl-4-vinylimidazole as a clear yellow oily residue: yield = 0.45 g, 25%. The reaction was then increased in scale to prepare 25 g of crude 2-methyl-4-vinylimidazole that was purified by crystallization from ethyl acetate. Thus, 10.5 g of crude 2-methyl-4-vinylimidazole was charged to a 250 mL Erlenmeyer flask and warmed to dissolve in a minimum amount of ethyl acetate. The solution was cooled, held at room temperature for 1 h, and subsequently stored in a refrigerator at 0 °C for 3 days. Needle-like white crystals formed and were collected by vacuum filtration: yield = 8.65 g, 82%—^1^H NMR (in methanol-d4) 2.38 (3H, s, −CH_3_); 6.50 (1H, m, vinyl H-C); 5.01 (1H, q, trans-vinyl H); 5.58 (1H, q, cis vinyl H); 6.97 (1H, s, C-5H).

#### 2.4.5. 1-Butyl-2-methyl-4-vinylimidazole **(5)**

Crystalline 2-methyl-4-vinylimidazole (2.15 g, 19.7 mmol) and tetrahydrofuran (20 mL) were charged to a 150 mL round-bottomed flask, equipped with y-tube, rubber stopper, reflux condenser with gas inlet, and magnetic stirring bar. The suspension was stirred under argon and cooled in an ice bath. Potassium *tert*-butoxide, 1M in tetrahydrofuran (39.5 mL, 39.5 mmol), was injected into the flask and the reaction mixture, which turned milky white and was left to stir for 15 min. n-Butyl bromide (2.55 mL, 23.7 mmol) was added drop-wise and the reaction mixture was held under Ar at ambient temperature and stirred overnight prior to the removal of the solvent in vacuo to yield a yellow oil that was dissolved in 50 mL of diethyl ether. The ether solution was washed with 100 mL of 5% aqueous hydrochloric acid and separated from the water layer. Potassium carbonate was added to the aqueous layer until the pH increased to 11. The aqueous layer was then extracted three times with 100 mL aliquots of diethyl ether. All ether solutions were combined and dried over magnesium sulfate. The ethereal solution was filtered and the solvent was removed via rotary evaporation to give a light-yellow oil: yield = 0.8 g, 25%.

Crude 1-butyl-2-methyl-4-vinylimidazole (1.50 g, 9.15 mmol), synthesized as described above at a larger scale, was charged to a 10 mL one-necked round bottom flask, equipped with a vacuum-jacketed, short path distillation head. The crude monomer was distilled in vacuo to yield 0.84 g (56%) of “pure” 1-butyl-2-methyl-4-vinylimidazole. The clear colorless product was stored at 0 °C: ^1^H NMR (in methanol-d4) 0.88 (3H, t, N−(CH_2_)_3_CH_3_); 1.35 (2H, m, N−CH_2_CH_2_CH_2_CH_3_); 1.62 (2H, m, N−CH_2_CH_2_CH_2_CH_3_); 3.73 (2H, t, N−CH_2_CH_2_CH_2_CH_3_); 2.32 (3H, s, −C-2CH_3_); 5.01 (1H, d/d, trans-vinyl H); 5.68 (1H, d/d, cis-vinyl H); 6.47 (1H, q, vinyl H−C); 6.70 (1H, s, C-5H). The ^1^H NMR spectrum shows the presence of a very small amount (less than 5%) of 1-butyl-2-methyl-5-vinylimidazole.

#### 2.4.6. 1-Butyl-2,3-dimethyl-4-vinylimidazolium Triflate **(6)**

Freshly distilled 1-butyl-2-methyl-4-vinylimidazole (0.84 g, 5.1 mmol), containing a small amount of 1-butyl-2-methyl-5-vinylimidazole and 8 mL dichloromethane, was charged to a 50 mL one-neck, round-bottom flask, equipped with y-tube, rubber serum cap, reflux condenser with gas inlet, and magnetic stir bar. The vessel was immersed in ice bath. Under an argon blanket, methyl trifluoromethanesulfonate (0.69 mL, 6.1 mmol), which was dissolved in 10 mL of dichloromethane, was added drop-wise by syringe. The reaction mixture was stirred at 0 °C for 2 h prior to removal of dichloromethane and unreacted methyl trifluoromethanesulfonate, in vacuo, at 0 °C, to provide a white crystalline product melting at 45 °C: yield = 1.68 g, 100%—^1^H NMR (in methanol-d4) 0.92 (3H, t, N−(CH_2_)_3_CH_3_); 1.35 (2H, m, N−CH_2_CH_2_CH_2_CH_3_); 1.75 (2H, m, N−CH_2_CH_2_CH_2_CH_3_); 4.09 (2H, t, N−CH_2_CH_2_CH_2_CH_3_); 2.54 (3H, s, −C-2CH_3_); 3.73 (3H, s, N−CH_3_); 5.53 (1H, d/d, cis-vinyl H); 5.83 (1H, d/d, trans-vinyl H); 6.60 (1H, m, vinyl H−C); 7.66 (1H, s, C-5H).

### 2.5. Polymerization of 1-Butyl-2,3-dimethyl-4-vinylimidazolium Triflate

1-Butyl-2,3-dimethyl-4-vinylimidazolium trifluoromethanesulfonate (1.68 g, 5.80 mmol) was dissolved in ethyl acetate (8 mL) and ethanol (2 mL) and charged to a polymerization tube at 0 °C. A 0.012 molar solution of AIBN in ethyl acetate (1 mL) was added to the polymerization tube, and the reaction mixture was degassed in three freeze–thaw cycles, the flame sealed, and immersed in a water bath at 65 °C for 20 h. A viscous polymer solution was formed. The polymerization tube was opened and the reaction mixture was precipitated in 300 mL of diethyl ether. The product was isolated by centrifugation and dried in an inert atmosphere to yield 1.68 g, 100%, of a fluffy white polymer, M_n_~39.6 kDa, polydispersity = 1.75.

### 2.6. Ion Exchange of Poly(1-Butyl-2,3-dimethyl-4-vinylimidazolium Triflate)

A “stock solution” with a concentration of 0.02 g/mL of poly(1-butyl-2,3-dimethyl-4-vinylimidazolium triflate) in methanol was used in the ion exchange process. Thus, 10 mL aliquots (~0.70 mmol) of stock solution was mixed with 10 mL aliquots (~0.90 mmol) of methanolic solutions of lithium trifluoromethylsulfonamide or ammonium hexafluorophosphate. The precipitate that formed immediately upon mixing the two solutions was isolated by centrifugation and washed repeatedly with methanol to yield poly(1-butyl-2,3-dimethyl-4-vinylimidazolium trifluoromethylsulfonylimide) and poly(1-butyl-2,3-dimethyl-4-vinylimidazolium hexafluorophosphate), respectively. In order to ensure that these polymers were fully ion-exchanged, the wet trifluoromethylsulfonylimide (TFSI) and hexafluorophosphate (PF_6_^−^) salts were washed with additional aliquots of methanolic solutions of lithium TFSI and ammonium PF_6_^−^. These wet precipitates were exhaustively washed with methanol to remove any excess TFSI^−^ or PF_6_^−^ salts. The efficiency of this process in affecting full ion exchange was validated by comparative ^19^F NMR of the PF_6_^−^ polymer and the pure, not ion-exchanged, CF_3_SO_3_^−^ polymer that showed no signal for the fluorine bound to a trifluoromethyl moiety in the ion-exchanged PF_6_^−^ polymer.

The ion-exchanged polymers were dried in vacuo prior to thermal analysis.

## 3. Results and Discussion

### 3.1. Synthesis and Ion Exchange of Poly(1-Butyl-2,3-dimethyl-4-vinylimidazolium Triflate)

The vinylimidazolium polymers, which have a hydrogen atom on the 2-position on the imidazolium ring, may be candidates for membrane electrolytes in capacitive electrochemical devices. However, in battery applications, the acidic hydrogen at the 2-position of the imidazole ring may react with electrons formed at the cathode, releasing hydrogen gas and causing irreversibility in charging–discharging cycles [41]. A solution to this potential problem is the synthesis of 2-methylimidazolium polymers with structures such as those shown in Figure 2. The family of 1-butyl-2,3-dimethyl-4-vinylimidazolium polymers with triflate (Tf^−^), hexafluorophosphate (PF_6_^−^) and trifluoromethylsulfonylimide (TFSI^−^) anions was synthesized using procedures described in detail in the Materials and Methods Section. Thus, freshly distilled, 1-butyl-2-methyl-4-vinylimidazole **(5)** was quaternized with methyltriflate to yield 1-butyl-2,3-dimethyl-4-vinylimidazolium triflate **(6)**. The entire reaction sequence, starting from 4(5)-vinylimidazole, **(1)** is outlined below in Figure 4.

Each of the reactions in the sequence is inherently high yield; however, the yields of tautomeric 2-methyl-4(5)-vinylimidazole **(4)** and 1-butyl-2-methyl-4-vinylimidazole **(5)** were adversely impacted by difficulties in isolating them from their reaction mixtures. The butylation of **(4)** was highly regiospecific, as evidenced by the fact that the ^1^H-NMR spectrum of **(5)** showed only a miniscule amount of 1-butyl-2-methyl-5-vinylimidazole. The quaternization with methyl triflate was very facile with **(5)** being quantitatively converted to **(6)**. The successful synthesis of **(6)**, 1-butyl-2,3-dimethyl-4-vinylimidazolium triflate, is confirmed by its ^1^H-NMR spectrum shown in Figure 5.

The chemical 1-Butyl-2,3-dimethyl-4-vinylimidazolium triflate was homopolymerized free-radically in ethyl acetate, initiating with AIBN, and was isolated and dried in accordance with the procedure detailed in the Materials and Methods Section. The poly(styrene) equivalent number average molecular weight was ~39.6 KDa; the polydispersity was 1.75. The ^1^H NMR spectrum of poly(1-butyl-2,3-dimethyl-4-vinylimidazolium triflate), P23DM4VIm^+^ Tf^−^, is shown in Figure 6. The extraneous peak at about 2.9 ppm corresponds to residual protons in the DMSO-d_6_.

As described in the Materials and Methods section, a methanolic solution of P23DM4VIm^+^ Tf^−^ was ion exchanged with methanolic solutions of trifluoromethylsulfonylimide, (TFSI^−^) and hexafluorophosphate, (PF_6_^−^) salts, respectively. The resultant P23DM4VIm^+^ TFSI^−^ and P23DM4VIm^+^ PF_6_^−^ salts precipitated from the mixed methanol solutions, and the precipitates were washed with methanolic solutions of TFSI^−^ and PF_6_^−^ salts to ensure full ion-exchange.

The efficiency of this process in affecting full ion exchange was validated by comparative ^19^F NMR of the PF_6_^−^ polymer and the pure, not-ion-exchanged CF_3_SO_3_^−^ polymer that showed no signal for the fluorine bound to a trifluoromethyl moiety in the ion-exchanged PF_6_^−^ polymer.

### 3.2. The Glass Transition in Ionic Polymers

In their ground-breaking study of the glass transition in several families of ionic polymers, Eisenberg et al. [45] concluded that the T_g_ of ionic polymers was determined by the magnitude of the cation–anion interaction and that T_g_ was directly proportional to the ratio of the charge on the ion and the separation between the ionic moiety tethered to the polymer and its counterion, *q*/*a*. This conclusion was supported by studies of phosphate, silicate, acrylate and aliphatic ionene polymer systems. [45,46] Eisenberg et al. found that the slope of plots of T_g_ versus *q*/*a* in these three polymer systems ranged from 625 to 730. Building on these works, Tsutsui and Tanaka [47] and Peiffer [48] found the glass transition temperature of ionic polymers to be dependent on the molar ionic cohesive energy, the universal gas constant, and a factor related to intramolecular interactions. They derived Equation (1), which fitted the T_g_ versus *q*/*a* data for phosphate, acrylate and aliphatic ionene polymer systems:(1)Tg=2NAe23CR(qa)+12R(25)
where *N_A_* is Avogadro’s number, e is the electronic charge, *C* is a constant, *q*/*a* is the ratio of the charge on the ion to the separation between the ion pair, and *R* is the universal gas constant. Given the glass transition data reported in the literature for 3-ethyl-1-vinylimidazolium polymers [12,27,33] and 3-butyl-1-vinylimidazolium polymers, [12] the relationship between T_g_ and *q*/*a* can be evaluated for vinylimidazolium polymers. These data, along with their associated molecular ion volumes [49] and ionic radii, are shown in Table 1.

When plotting the data from Table 1, T (K) versus the ratio of q (in units of 1 electron) over *a* (in Angströms), one sees (Figure 7) a linear dependence of T_g_ on *q*/*a* with a slope of 608. Thus, the glass transition temperatures of poly(ionic liquids) derived from 1-vinylimidazolium polymers appear to be linearly related to the ratio of the charge on the ion and the separation between the ionic moiety tethered to the polymer and its counterion, *q*/*a*. Moreover, the slope of the plot is close to those reported in the historic literature by Eisenberg, et al. for phosphate, silicate, acrylate and aliphatic ionene polymer systems [45,46].

The molecular volume of PEVIm^+^ is taken to be 0.156 ± 0.018 nm^3^, equal to that reported for EMIm^+^ [49]. The molecular volume of PBVIm^+^ is taken to be 0.196 ± 0.0.021 nm^3^, equal to that reported for C4MIm^+^ [49]. Assuming anions and cations to be spherical, ionic radii of the respective ions and respective anions and cation were calculated from the molecular volume of ions [50]. The distance between the anion and the cation, *a*, was calculated as the sum of the radii of the respective anions and cations.

### 3.3. The Glass Transition in P23DM4VIm^+^ Salts

In the present work, the glass transition characteristics of P23DM4VIm^+^ Tf^−^, PF_6_^−^, and TFSI^−^ salts were evaluated by differential scanning calorimetry. The thermograms for these salts are similar to each other (see Appendix A) with the heat capacity change around the glass transition temperature for all salts in this group of 2-methylimidazolium polymers exhibiting an “excess enthalpy” peak, often seen as a result of densification, in aged polymer glasses [51]. Enthalpic relaxation peaks are also often seen when materials are rapidly cooled to a temperature far below their T_g_ [52]. In the heating cycle, the T_g_-mid of the triflate salt is 371 K (98 °C). T_g_-mid of the trifluoromethylsulfonylimide salt is 372 K (99 °C), nearly identical to that of the triflate salt. The heating cycle glass transition temperature for the hexafluorophosphate salt is 357 K (84 °C). The glass transition temperatures of all three polymers are, therefore, very close to each other. In light of the strong dependence of the T_g_ of 1-vinylimidazolium polymers (PEVIm^+^ and PBVIm^+^) on the distance separating the ion pair, evidenced in Figure 7, and the substantial differences in the glass transition temperatures of the corresponding P4VIm^+^ salts [33], the proximity in the glass transition temperatures of the P23DM4VIm^+^ salts is surprising. The mid-point glass transition temperatures of P23DM4VIm^+^ salts derived from 1-butyl-2,3-dimethyl-4-vinylimidazolium polymers, and those of the corresponding P4VIm^+^ salts derived from 1-ethyl-3-methyl-4-vinylimidazolium polymers, are displayed in Table 2. Pertinent information on the molecular volume of anions and imidazolium cations and the distance separating the ion pairs is also included in Table 2.

Using the data in Table 2, the glass transition temperatures of PF_6_^−^, Tf^−^, and TFSI ^−^ salts of P23DM4VIm^+^ and P4VIm^+^ polymer salts are plotted below (see Figure 8) as a function of the ratio of the charge and the separation between the ionic moieties on the polymer.

The molecular volume of 23DM4VIm^+^ is assumed to be 0.229 ± 0.012 nm^3^, equal to that reported for C4MMIm^+^ [49]. The molecular volume of 4VIm^+^ is assumed to be 0.156 ± 0.018 nm^3^, equal to that reported for EMIm^+^ [49]. Assuming anions and cations to be spherical, the ionic radii of the respective ions and respective anions and cation were calculated from the molecular volume of ions [50]. The distance between the anion and the cation, a, was calculated as the sum of the radii of the respective anions and cations.

As was observed with PEVIm^+^ and PBVIm^+^ salts, the P4VIm^+^ salts (see Figure 8) also show a strong dependence of the glass transition temperature on anion size and q/a. The line in the plot of the data in Table 2 for poly(1-ethy-3-methyl-4-vinylimidazolium) salts is the fit to a second-order exponential. It was demonstrated in our earlier work [33] that the T_g_ of P4VIm^+^PF_6_^−^ is elevated by intramolecular bridging between imidazolium residues positioned 1,3- or 1,5- along the P4VIm^+^ chain. Accordingly, the non-linearity of the plot for P4VIm^+^ salts (Figure 8) as compared to that for PEVIm^+^ and PBVIm^+^ salts (Figure 7) is not unexpected.

In stark contrast to poly(vinylimidazolium salts) with a hydrogen at the 2-position of the imidazolium ring, the glass transition temperatures of Tf^−^, TFSI^−^ and PF_6_^−^ salts from P23D4VIm^+^ salts are substantially invariant, independent of the size of the anion. The glass transition temperature of polymers in this set falls within a mere 15-degree temperature range, 357–372 K.

Figure 9 shows space-filling molecular renderings of 1-butyl-2,3-dimethyl-4-vinylimidazolium triflate (Figure 9B) and 1-ethyl-3-methyl-4-vinylimidazolium triflate polymer triads (Figure 9A) that were created using the 3-D rendering tool in ACD ChemSketch (ACD, Toronto, ON, Canada).

These molecular renderings serve to illustrate that the proximity of the triflate anion to the imidazolium group is more hindered in the triad of P23D4VIm^+^ (Figure 9B), wherein the 2-methyl substituent forces the triflate anion to be further removed from imidazolium moiety. On the basis of data in Table 2, one would estimate that the minimum distance separating the P23DM4VIm^+^ cation and the triflate anion is 6.95 Å, while that separating the P4VIm^+^ cation and the triflate anion is decreased to 6.77 Å. Given the substantive difference in the glass transition temperature of the triflate salts of P4VIm^+^ and P23DM4VIm^+^, the methyl group at the 2-position of the imidazolium ring must be having a greater influence than simply forcing a 26% increase in the steric separation between the cation and anion. Fernandes et al. [53] carried out a systematic study on the relative interaction energies of ionic liquids. These authors report that while cation–anion interactions in ionic liquids are dominated by coulombic forces (the distance separating the ion pair), hydrogen bonding also plays an important role. For imidazolium-based ionic liquids the contribution of hydrogen bonding between the cation and the anion is determined by the acidic hydrogen in the 2-position of the imidazolium ring. In a work by Kirchner [54], this was quantified by comparing the interaction energies of 1-butyl-3-methylimidazolium trifluoromethylsulfonylimide (BMIm^+^TFSI^−^) and 1-butyl-2,3-dimethylimidazolium trifluoromethylsulfonylimide (DMIm^+^TFSI^−^), wherein an energetic difference between the two salts of 20 kJmol^−1^ was calculated. The higher dissociation energy for the separation of the cation in BMIm^+^TFSI^−^ as compared to DMIm^+^TFSI^−^ was also observed experimentally in the work of Fernandes et al. [53], with values of relative interaction energies of 0.491 and 0.426 eV, respectively, being reported. It is thus reasonable to attribute the invariance in the glass transition of P23D4VIm^+^ with different anions to a combination of minimal H-bonding interactions between the respective anions and the 23D4VIm^+^ moiety and steric separation enforced by having substituents on the 1, 2, 3, and 4 positions of the imidazolium ring.

## 4. Summary and Conclusions

The chemical 1-Butyl-2,3-dimethyl-4-vinylimidazolium triflate (2,3DM4VIm^+^Tf^−^) was synthesized in a pure, dry (water-free) state by direct alkylation of 1-butyl-2-methyl-4-vinylimidazole in CH_2_Cl_2_ with methyl triflate at 0 °C. Moreover, 2,3DM4VIm^+^Tf^−^ was polymerized with a free-radical initiator, and the resulting polymer was ion exchanged to create PF_6_^−^, and TFSI^−^ salts. This may be the first literature report of the synthesis and polymerization of an ionic liquid vinylimidazolium monomer with a substituent at the 2-position of the imidazolium ring. The glass transition characteristics of TFSI^−^, PF_6_^−^ and triflate salts of the tetrasubstituted 2,3DM4VIm^+^ polymers are anomalous as compared to those of corresponding TFSI^−^, PF_6_^−^ and triflate salts of trisubstituted 4-vinylimidazolium polymers without a methyl substituent at the 2-position. The glass transition temperatures for the P4VIm^+^ salts vary from 88 °C for the TFSI^−^ salt to 153 °C for the CF_3_SO_3_^−^ salt and 200 °C for the PF_6_^−^ salt. Those for the corresponding P23DM4VIm^+^ salts are 99 °C, 98 °C and 84 °C, respectively.

The invariance in the glass transition of P23DM4VIm^+^ with different anions is attributed to a combination of the steric separation enforced by having substituents on the 1-, 2-, 3-, and 4- positions of the imidazolium moiety. This dramatically diminishes the contribution of the counterion size to the equilibrium distance between the center of the anion and cation and minimal H-bonding interactions in the 2-substituted imidazolium polymer salts. It is believed that this is the first observation of the substantial independence of the glass transition of an ionic polymer on the nature of its counterion. One might even speculate that the interaction between the ion pairs may be reduced to the point that the onset of motion of the polymer backbone is effectively decoupled from counterion motion.

## Figures and Tables

**Figure 1 polymers-14-00254-f001:**
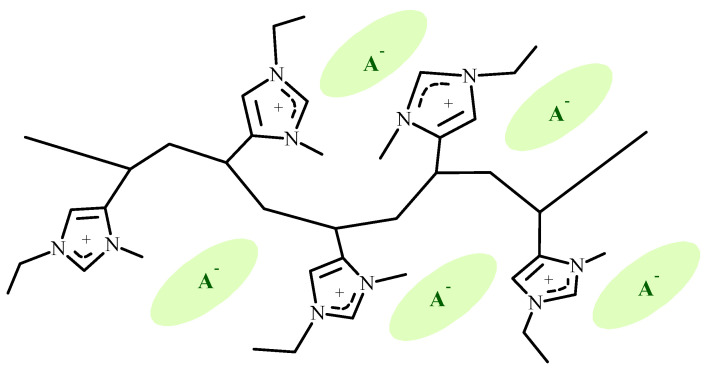
Pentad segment of a 1-ethyl-3-methyl-4-vinylimidazolium polymer.

**Figure 2 polymers-14-00254-f002:**
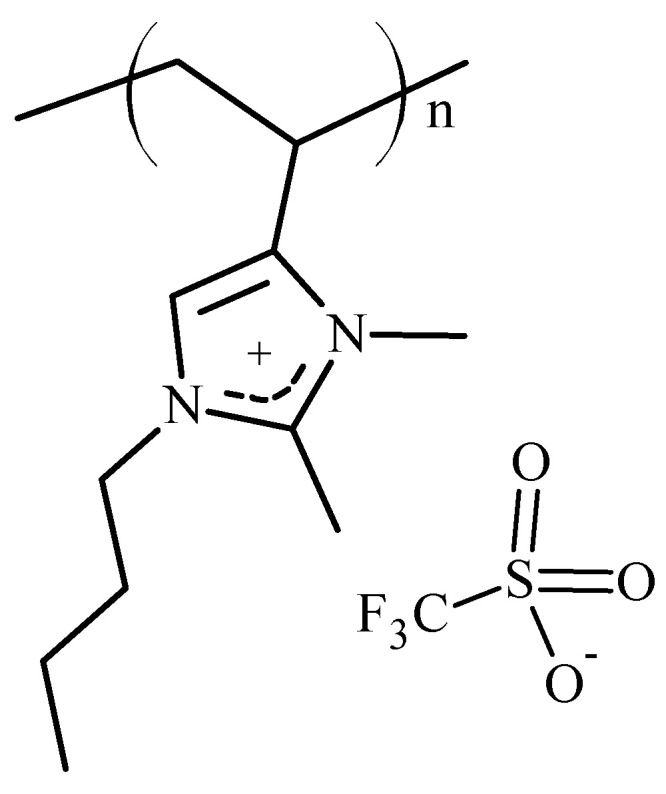
Structure of poly(1-butyl-2,3-dimethyl-4-vinylimidazolium triflate).

**Figure 3 polymers-14-00254-f003:**
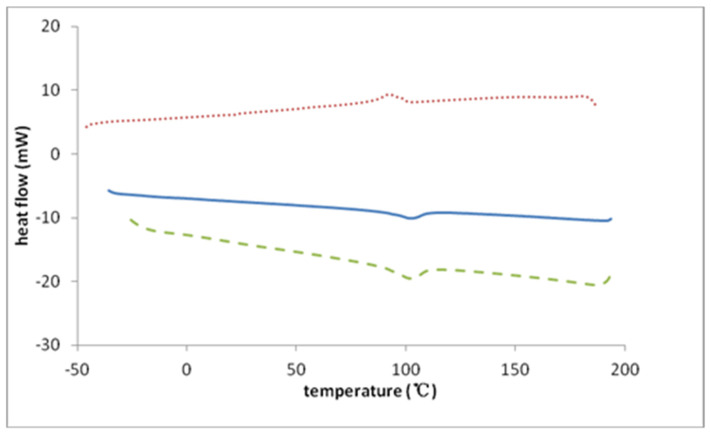
DSC scans for P23DM4VIm^+^Tf^−^: solid blue line (20 °C/min—heating cycle), dotted red line (20 °C/min—cooling cycle), dashed green line (40 °C/min—rapid heating cycle). Endothermic heat flow is indicated by downward deflection from the baseline.

**Figure 4 polymers-14-00254-f004:**
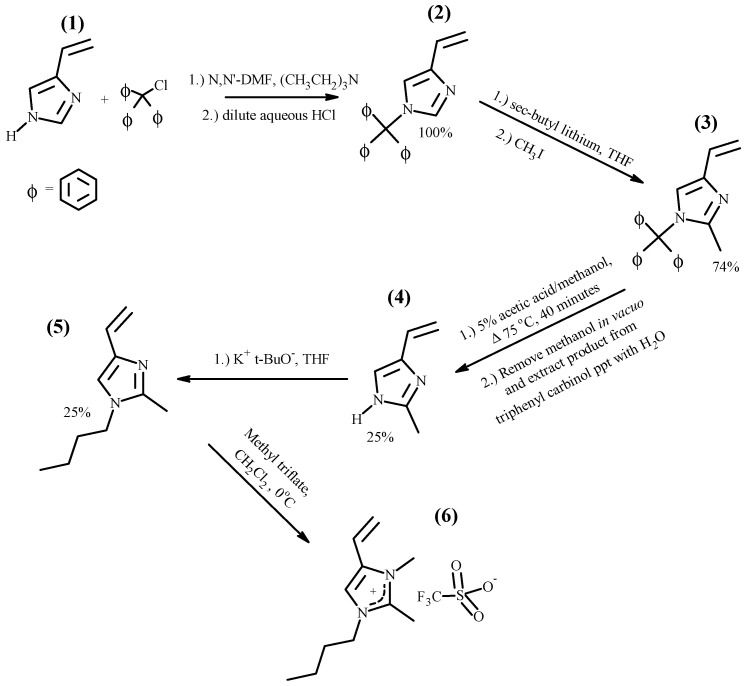
Synthesis of 1-butyl-2,3-dimethyl-4-vinylimidazolium triflate.

**Figure 5 polymers-14-00254-f005:**
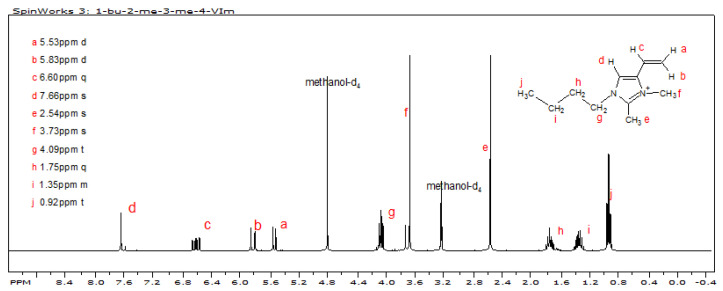
^1^H NMR spectrum of 1-butyl-2,3-dimethyl-4-vinylimidazolium triflate.

**Figure 6 polymers-14-00254-f006:**
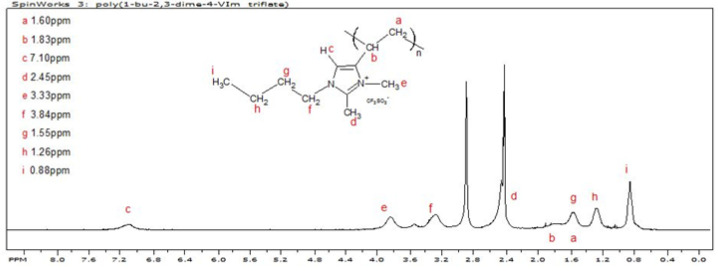
^1^H NMR spectrum of poly(1-butyl-2,3-dimethyl-4-vinylimidazolium triflate).

**Figure 7 polymers-14-00254-f007:**
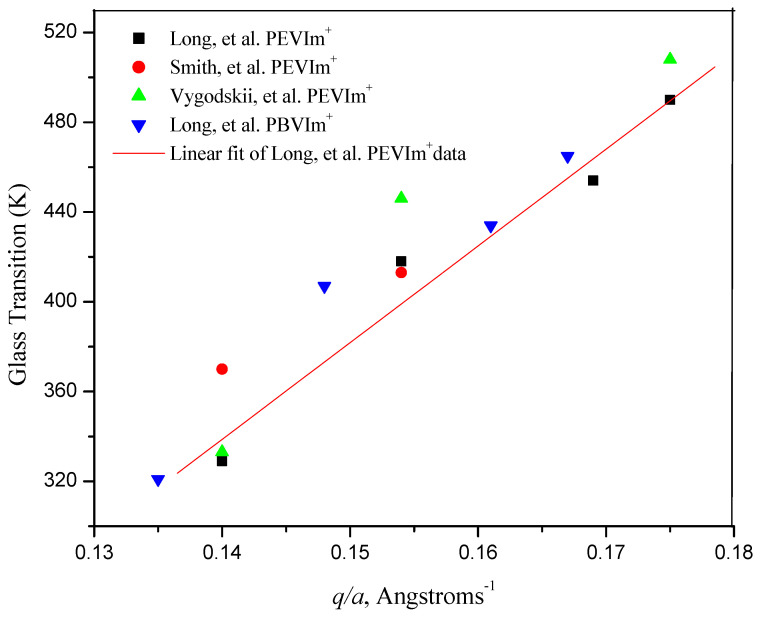
Glass transition temperature (K) versus the ratio of the charge on the ion and the separation between the ion pair, *q*/*a*, for poly(3-ethy-1-vinylimidazolium) and poly(3-butyl-1-vinylimidazolium) salts. (■ and▲ Long et al. [12]; ● Smith et al. [33]; ▲ Vygodskii et al. [27]).

**Figure 8 polymers-14-00254-f008:**
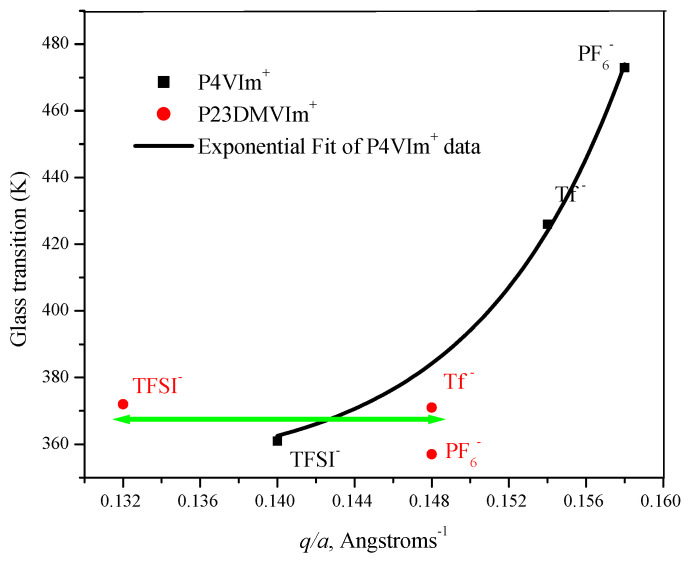
Glass transition temperature (K) versus the ratio of the charge on the ion and the separation between the ion pair, *q*/*a*, for poly(1-ethyl-3-methyl-4-vinylimidazolium) and poly1-butyl-2,3-dimethyl-4-vinylimidazolium) salts.

**Figure 9 polymers-14-00254-f009:**
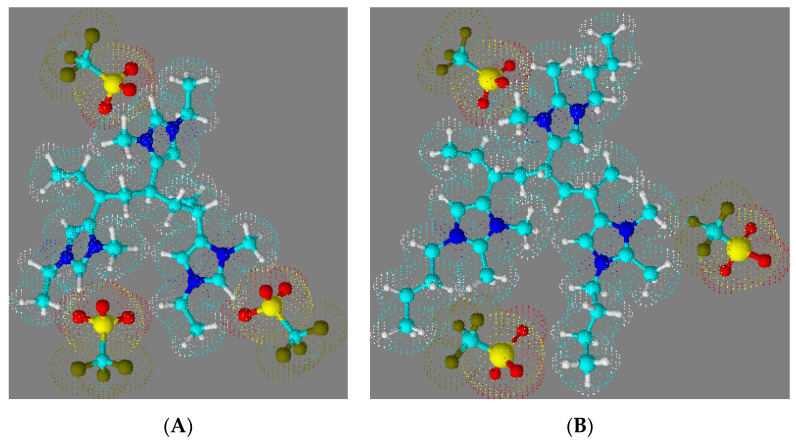
Space-filling molecular renderings of (**A**) 1-ethyl-3-methyl-4-vinylimidazolium triflate and (**B**) 1-butyl-2,3-dimethyl-4-vinylimidazolium triflate polymer triads.

**Table 1 polymers-14-00254-t001:** T_g_ of Poly(3-ethyl-1-vinylimidazolium) and Poly(3-butyl-1-vinylimidazolium) salts. [12].

Anion (Volume, nm^3^) [46]								
Polymer Composition	Br^−^		BF_4_^−^		Tf^−^		TFSI^−^	
	(0.054 ± 0.007)		(0.073 ± 0.009)		(0.131 ± 0.015)		(0.232 ± 0.015)	
	PEVIm^+^	PBVIm^+^	PEVIm^+^	PBVIm^+^	PEVIm^+^	PBVIm^+^	PEVIm^+^	PBVIm^+^
T_g_, K	490 508 [27]	465	454	421	418 413 [29]	407	329 370 [29] 333 [27]	321
*a*, Å	5.71	5.97	5.93	6.19	6.49	6.75	7.15	7.41
*q*/*a* *	0.175	0.167	0.169	0.161	0.154	0.148	0.140	0.135

* The quantity, *q*/*a*, is the ratio of the charge on the ion and the separation between the ionic moiety tethered to the polymer and its counterion.

**Table 2 polymers-14-00254-t002:** Poly(1-butyl-2,3-dimethyl-4-vinylimidazolium) and poly(1-ethyl-3-methyl-4-vinylimidazolium) salts.

Polymer Composition	Tf^−^		TFSI^−^		PF_6_^−^	
	(0.131 ± 0.015)		(0.232 ± 0.015)		(0.109 ± 0.008)	
	P23D4VIm^+^	P4VIm^+^	P23D4VIm^+^	P4VIm^+^	P23D4VIm^+^	P4VIm^+^
T_g_, K	371	426 [29]	372	361 [29]	357	473 [29]
*a*, Å	6.95	6.40	7.60	7.15	6.77	6.32
*q*/*a* *	0.148	0.154	0.132	0.140	0.148	0.158

* The quantity, *q*/*a*, is the ratio of the charge on the ion and the separation between the ionic moiety tethered to the polymer and its counterion.

## Data Availability

Data supporting reported results can be found through https://scholarworks.rit.edu (accessed on 1 December 2021), in Darren Smith “Synthesis and characterization of poly(ionic liquids) derived from 1-ethyl-3-methyl-4-vinylimidazolium triflate” (2011); Fan Yang “Synthesis and Characterization of Ionic Liquid Monomers and Polymers Derived from 2-substituted-1,3-dialkyl-4(5)-vinylimidazolium salts” (2012); and the RIT Ph.D. Dissertation of Meng Zhao “Thermal Analysis and Dielectric Spectral Characteristics of Poly(ionic Liquids): Towards exploration of their utility in capacitive electrochemical devices” (2014).

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
