# Peer review of "Anomalous Thermal Characteristics of Poly(ionic liquids) Derived from 1-Butyl-2,3-dimethyl-4-vinylimidazolium Salts"

_polymers, 2022, doi:10.3390/polym14020254_

Round 1

Reviewer 1 Report

In their submission to Polymers entitled "Anomalous Thermal Characteristics of Poly(ionic liquids) Derived from 1-Butyl-2,3-Dimethyl-4-Vinylimidazolium Salts", Thomas W. Smith and coworkers report on the synthesis of the ionic liquid 1-butyl-2,3-dimethyl-4-vinylimidazolium triflate, and its subsequent polymerization, and ion exchange to prepare 1-butyl-2,3-dimethyl-4-vinylimidazolium polymeric materials. The manuscript may be publishable, but it should be reviewed after major revisions. This is a short and interesting approach, but there are some points to be addressed before publication. However, the novelty in terms of polymer chemistry is insufficient for Polymers. Moreover, the references are quite unbalanced and confusing. 
Please also check the following points:
1) The experimental part should describe in enough detail, including the spectroscopic characterization (1H NMR peak assignement ), melting points of compounds 1-3. 
2) Yields should be given with no decimals.
3) The quality of Figure 3 andd Figure 6 needs to be improved.
4) In Figure 5, the intregration of the peaks should be included.

5) A revison of the English style is also recommended.

Reviewer 2 Report

In this paper, the authors describe the synthesis of 1-Butyl-2,3-dimethyl-4-vinylimidazolium triflate and its polymerization. The counterion of resulting poly(ionic liquid) was also replaced by hexafluorophosphate or trifluoromethylsulfonylimide anions through ion exchange process. The glass transition of these polymers with the different counterions, was found to be without much difference. This is a unique result for poly(ionic liquids) and the authors attributed this to the steric separation of the ions and minimizing the H-bonding interactions of the 2-substituted imidazolium salts of the polymers. The results described in this paper are very interesting. The paper is well written and organized. This reviewer recommends publication of the paper after the following minor revisions:

  • The authors mentioned that ion exchange was confirmed by 19F-NMR. The spectra should be added to the Supporting information.
  • The 13C-NMR spectra of the monomer and the polymers should be added.

Round 2

Reviewer 1 Report

The authors have addressed all the issues suggested bu the reviewers and have improved the quality of the manuscript. Therefore, it can be published at this stage.

Reviewer 2 Report

The authors have revised their paper while addressing properly all comments of this reviewer.